# Coherence protection of spin qubits in hexagonal boron nitride

Andrew J. Ramsay ®[1], Reza Hekmati[2], Charlie J. Patrickson ®[3], Simon Baber ®[3], David R. M. Arvidsson-Shukur ®[1], Anthony J. Bennett[2,4] & Isaac J. Luxmoore ®[3] ✉

Spin defects in foils of hexagonal boron nitride are an attractive platform for magnetic field imaging, since the probe can be placed in close proximity to the target. However, as a III-V material the electron spin coherence is limited by the nuclear spin environment, with spin echo coherence times of ∽100 ns at room temperature accessible magnetic fields. We use a strong continuous microwave drive with a modulation in order to stabilize a Rabi oscillation, extending the coherence time up to ∽4μs, which is close to the 10 μs electron spin lifetime in our sample. We then define a protected qubit basis, and show full control of the protected qubit. The coherence times of a superposition of the protected qubit can be as high as 0.8 μs. This work establishes that boron vacancies in hexagonal boron nitride can have electron spin coherence times that are competitive with typical nitrogen vacancy centres in small nanodiamonds under ambient conditions.

Hexagonal boron nitride (hBN) is a wide-bandgap Van der Waals material used as an insulator in two-dimensional electronic devices[1]. Recently, there has been a growing interest in optically detected magnetic resonance (ODMR) experiments on spin defects. There are several reasons why. Firstly, ODMR provides a useful tool for identifying defects, and has been instrumental in identifying the boron vacancy in hBN[2,3]. Secondly, a good magnetic field sensor or spin-photon interface[4] requires a defect with both good optical and spin properties. There is mounting evidence that some hBN defects have excellent optical properties, with high brightness of up to 87% quantum efficiency[5] at visible wavelengths well-matched to silicon detectors[6], and suggestions of transform-limited transitions with a high fraction of emission occurring through the zero-phonon line[7,8]. Thirdly, as a two-dimensional material it may be possible to make magnetic-field sensing foils that allow the spin-defect to be placed in close-proximity to the target of interest[9–11]. In the leading platform, nanodiamonds, the spin coherence properties are adversely affected by the surface states, typically limiting spin echo coherence times to a few μs for diameters of a few nm[12].

In hBN, there are a few recent studies on single bright ODMR-active defects[13–15], which are possibly carbon related. But so far, most ODMR work in hBN has focused on ensembles of boron vacancies[2,9,16–19], because although they suffer low brightness, they are easy to generate[19], their internal energy levels have been theorized[18,20,21] and allow facile spin-pumping by a green laser.

In any III-V material the nuclear spin environment is expected to limit the electron spin coherence times. So far, in hBN there have been claims of spin echo times in the μs-regime, at room temperature and moderate magnetic-field[16,22], which is competitive with negatively charged nitrogen vacancy (NV)-centres in small nano-diamonds[12]. However, this has been challenged by experiments that show sub-100 ns spin echo times in isotopically purified material, which are further supported by calculations of the decoherence expected from the electron-nuclear interactions[3]. Since nuclear spin dephasing is largely a property of the host material[23], the boron vacancy serves as a model spin system to evaluate the potential of hBN as a host.

Here, we present a study of the spin properties of ensembles of negatively charged boron vacancies ($V_B^-$) in hBN. We confirm the findings of Haykal et al.[3] that the majority of the spin echo coherence is lost in $T_{echo} < 100$ ns at room temperature and milli-Tesla magnetic fields. However, we propose and demonstrate a solution. We show that by using a strong continuous microwave field, a method often referred

[1]Hitachi Cambridge Laboratory, Hitachi Europe Ltd., CB3 0HE Cambridge, UK. [2]School of Physics and Astronomy, Cardiff University, Queen's Building, CF24 3AA Cardiff, UK. [3]Department of Engineering, University of Exeter, EX4 4QF Exeter, UK. [4]School of Engineering, Cardiff University, Queen's Building, CF24 3AA Cardiff, UK. ✉e-mail: i.j.luxmoore@exeter.ac.uk

to as continuous concatenated dynamic decoupling (CCD), we can define a protected qubit basis with favourable coherence properties, allowing the Rabi oscillation damping time to be extended up to 4 μs. This method has previously been used in NV-centres in diamond[24-31], but here we show that it works extremely well in a III-V material with hostile nuclear environment, by using a Rabi drive stronger than the hyperfine coupling. Furthermore, we define a protected qubit basis in terms of an electron spin that rotates in-phase or out-of-phase with the expected Rabi oscillation. We then demonstrate full control of this protected qubit to show coherence times, $T_{pRabi}$, of up to 0.8 μs. The method avoids the use of cryogenic superconducting magnets, and is therefore suitable for room temperature applications. Whilst the boron vacancy may not ultimately be suitable for single spin applications, due to its low quantum efficiency and broad photoluminescence spectrum[17,18], ODMR on other single spin defects has been reported[13-15]. The CCD method presented here should be insensitive to the details of the nuclear bath, and therefore applicable in general to spin defects in hBN, and to other III-V materials with harsh nuclear spin environments.

## Results

### Boron vacancy optically detected magnetic resonance

Our device consists of flakes of hBN placed on top of a co-planar waveguide (CPW) fabricated on a sapphire substrate[17]. Following the recipe of ref. [19], boron vacancies are generated by carbon-ion irradiation at 10 keV and a dose of $1 \times 10^{14} cm^{-2}$. A DC magnetic field of 20 mT is applied along the c-axis of the hBN flakes, and an AC field is applied in-plane through the CPW. The coupling is sufficient to achieve Rabi frequencies in excess of 100 MHz. The sample is located beneath a microscope at room temperature and in air. Photoluminescence (PL) is

excited using a 532 nm laser, modulated using an acousto-optic modulator, and detected with a Si-APD module.

Only $V_B^-$ is expected to be optically active in the range of our detection system[32]. A sketch of the optical pumping cycle for optically detected magnetic resonance (ODMR) is shown in Fig. 1a. The $V_B^-$ has two unpaired electrons in an $S = 1$ triplet ground state. Excitation with a 532 nm laser preferentially optically pumps the $V_B^-$ into the $m_s = 0$ state, whose PL is slightly brighter than the $m_s = \pm 1$ states, allowing ODMR detection of the electron spin resonance. The PL is broadband and centred at ~850 nm, see Fig. 1b.

The Hamiltonian of the crystal ground-state can be expressed as $H = H_e + H_n + H_{en}$. The electron Hamiltonian,

$$H_e = DS_z^2 + E\left(S_x^2 - S_y^2\right) + \gamma_e B_z S_z, \qquad (1)$$

is composed of a zero-field splitting with $D = 3.479$ GHz, and $E = 59$ MHz (see Supplementary Fig. 2) and an electron Zeeman term with gyromagnetic ratio $\gamma_e \approx 28$ MHz/mT. $S_j$ are the $S = 1$ electron spin operators. D is consistent with previous works[2]. E is relatively large, perhaps due to high level of strain caused by use of carbon-irradiation to generate the defects, and is consistent with reports for ion-implanted samples[33]. The nuclear spin Hamiltonian is

$$H_n = \sum_k \gamma_n^k B_z I_z^k + \mathbf{I^k Q^k I^k}. \qquad (2)$$

$\mathbf{I^k}$ is the nuclear spin of nuclei $k$. The first term is a nuclear Zeeman term, and $\mathbf{Q^k}$ is a quadrupolar tensor. $H_{en}$ is the electron-nuclear

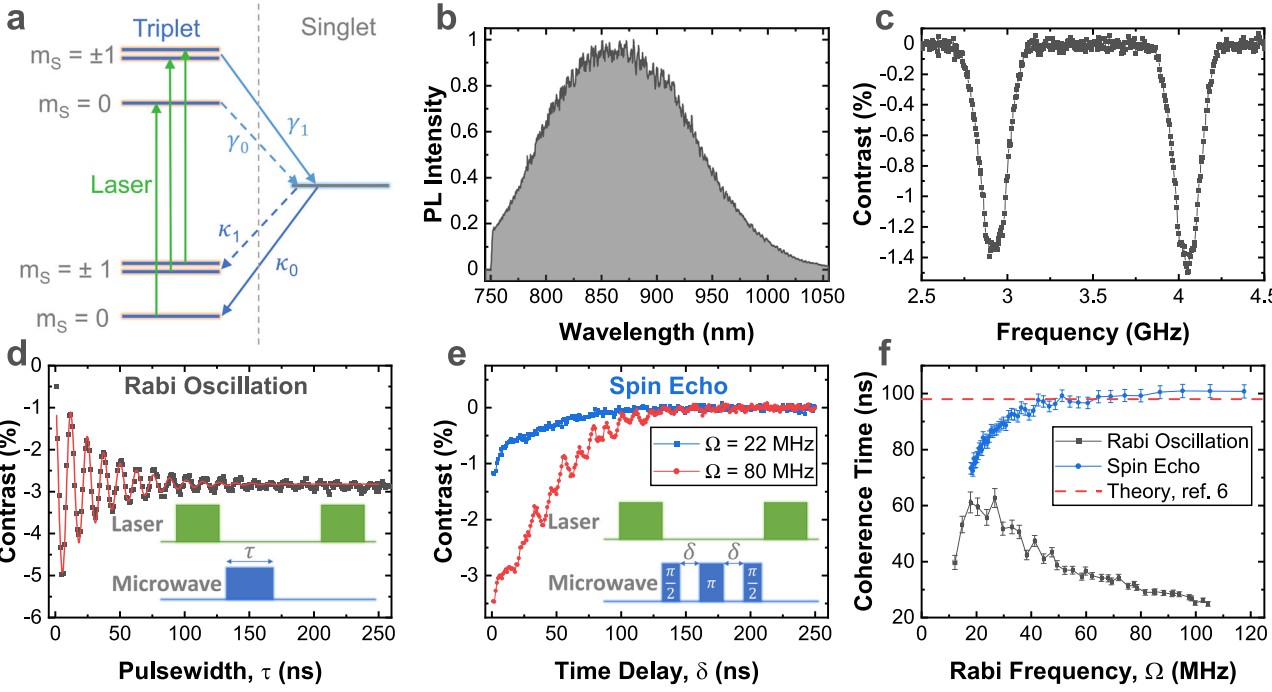

**Fig. 1 | Coherent control of unprotected boron vacancy spins. a** Energy levels involved in ODMR of the boron vacancy in hBN. Optical pumping combined with spin dependent inter-system crossing, initializes the system in $m_s = 0$ triplet ground state. This is slightly brighter than the $m_s = \pm 1$ states, allowing optical detection of the spin state. **b** Photoluminescence spectrum under 532 nm excitation. **c** Continuous wave ODMR spectrum, the FWHM of 150 MHz is mostly due to unresolved hyperfine structure. **d** Rabi oscillation of the $m_s = 0$ to $m_s = +1$ ground state transition **e** Comparison of spin echo at low (22 MHz, black) and high (80 MHz, red) Rabi frequency, Ω. At high Rabi frequency, the contrast is stronger since a higher proportion of spins are controlled. The insets to **d** and **e** show the pulse sequences used for Rabi oscillation and spin echo measurements,

respectively. **f** Comparison of coherence times extracted from Rabi oscillation (grey squares) and spin echo measurements (blue circles), as a function of Rabi frequency. For Rabi oscillations the coherence time decreases at high Rabi frequency due to fluctuations in drive strength[34]. For spin echo, the coherence time saturates at high Rabi frequencies, where the pulse is shorter relative to the spin precession time. The saturation value is close to that calculated by Haykal et al. (dashed-line) for hBN with natural isotope distribution[3]. The Rabi oscillations are fitted to $f(\tau) = a + b e^{-\tau/T_{Rabi}} \cos(\Omega\tau)$, and spin echo to $g(\delta) = -a \cdot e^{-2\delta/T_{echo}}$. The error bars indicate the standard error of the fit coherence times. Source data are provided as a Source Data file.

hyperfine interaction.

$$H_{en} = \sum_k \mathbf{S} \mathbf{A}^k \mathbf{I}^k \tag{3}$$

It is dominated by the three nearest neighbour nitrogen atoms with $A^{nn} = 47$ MHz, and $I(^{14}N) = 1^2$. The next nearest neighbour interactions have been calculated to be up to 6.8 MHz in ref. [21]. In our CW-ODMR measurements, see Fig. 1c, the hyperfine structure is not resolved, but the FWHM is similar to $3A$, corresponding to three nearest neighbours with $I = 1$.

## Unprotected electron spin

To start our investigation of the spin properties of $V_B^-$ we measure a Rabi oscillation, see Fig. 1d. The Rabi oscillation is sensitive to all inhomogeneities and noise sources, and has a coherence time of $T_{Rabi} < 60$ ns, which depends on the microwave power (Fig. 1f). At high microwave power, the dephasing rate is proportional to the Rabi frequency possibly due to fluctuations in the power of the microwave source. At low microwave power, the Rabi damping is dominated by fluctuations in the detuning[34].

To evaluate the intrinsic coherence times, we perform spin-echo measurements, see Fig. 1e. In principle, the measurement is insensitive to low frequency variations in the detuning, and errors in the pulse-area. However, at a moderate magnetic field of $B = 20$ mT, we find a $T_{echo} < 100$ ns, that saturates at high Rabi frequencies with fast pulses. We note that the $T_{echo}$ is comparable to the $T_{Rabi}$ times. These numbers are similar to those of Haykal et al.[3], who report that $T_{echo}$ is sensitive to $^{11}B$ content, and therefore limited by nuclear spins. In the high microwave-power regime our $T_{echo} = 100$ ns is a close match to their calculations for a natural 20:80 mix of $^{10}B$ and $^{11}B$ isotopes. We note that data reported by Liu et al.[35] is compatible with $T_{echo}$ of < 100 ns under similar conditions. In refs. [16,22] a longer μs-scale $T_{echo}$ was reported. However, these $T_{echo}$ were extracted from low contrast data for times larger than 200 ns. We could not observe a long-lived tail in the spin echo (see Supplementary Note 4 and Supplementary Fig. 3). At cryogenic temperature and few Tesla magnetic fields, $T_{echo} = 15$ μs has been reported[36].

## Stabilized Rabi oscillation

An important question for quantum devices based on hBN spin defects is whether one can create a protected qubit that is isolated from the nuclear spins, and thereby access a longer electron spin coherence. There exists an extensive literature on how to mitigate the impact of nuclear spins in III-V quantum dots[37–39], and for defects in group-IV materials[24,29,40,41]. For the boron vacancy, the electron-nuclear interaction is dominated by the three nearest nitrogen atoms and a strong hyperfine interaction of ~47 MHz. This few nuclei situation, contrasts with the case of a GaAs quantum dot, where a few thousand nuclei provide a bath of nuclear spins. It also contrasts with the case of group-IV materials, where the majority of nuclei have no nuclear spin, and electron spin dephasing due to hyperfine coupling is relatively weak.

To this end, we trial the use of a strong microwave field to create a protected spin-qubit subspace, and use concatenated continuous driving methods to coherently control the protected qubit[24,29] (see Supplementary Notes 1–3). This involves applying a continuous AC magnetic field along the x-direction to give a control Hamiltonian of the form:

$$H_c(t) = [\Omega \cos(\omega t + \phi) + 2\epsilon_m \sin(\omega t + \phi)\sin(\omega_m t - \theta_m)]S_x \tag{4}$$

The first term is the usual Rabi drive. The second term adds an amplitude modulated field in quadrature with the Rabi drive, and when $\omega_m = \Omega$ this acts to stabilize the Rabi oscillation. This can be

understood by considering only the $m_s = 0$, $m_s = +1$ electron spin states near resonance with the Rabi drive, and switching to the first rotating frame of the Rabi drive: $H'_c = e^{i\omega t\sigma_z/2}H_c e^{-i\omega t\sigma_z/2}$. In the case $\phi = 0$,

$$H'_c = \frac{1}{2}[\Omega\sigma'_x - 2\epsilon_m \sin(\omega_m t - \theta_m)\sigma'_y] \tag{5}$$

where $\sigma'_\alpha$ are the Pauli spin-1/2 matrices, and the superscript identifies the frame. The counter-rotating term has been neglected. The $\Omega$-term drives a Rabi oscillation in the y'z'-plane. The $\epsilon_m$ term applies a corrective rotation about the y'-axis, at a clock frequency of $\omega_m$.

Making a second rotating frame approximation to rotate into the dressed states basis, $H''_c = e^{i\omega_m t\sigma'_x/2}H'_c e^{-i\omega_m t\sigma'_x/2} - \omega_m\sigma'_x/2$.

$$H''_c = \frac{1}{2}(\Omega - \omega_m) + \frac{\epsilon_m}{2}[\sin(\theta_m)[-\sin(\phi)\sigma''_x + \cos(\phi)\sigma''_y] + \cos(\theta_m)\sigma''_z] \tag{6}$$

In the second rotating frame, the control field is equivalent to a DC magnetic field of magnitude $\epsilon_m$ that is oriented using the phases of the drive $\theta_m$ and $\phi$. By setting the drive such that $\Omega = \omega_m$, $\theta_m = \phi = 0$, the Hamiltonian reduces to $H''_C = \frac{1}{2}\epsilon_m\sigma''_z$. In this frame, an error in the Rabi frequency $(\Omega - \omega_m)$ results in a rotation about the x''-axis, and a spread in Rabi frequencies leads to a divergence in the rotation angle, and a damping of the unprotected Rabi oscillation. However, if an $\epsilon_m$-drive is applied, an error in the Rabi frequency results in a small tilt in the effective magnetic field, limiting the spread in the Bloch vector, resulting in a long-lived Rabi oscillation.

For a spin with no drive, the qubit is protected from bit-flip errors by an energy gap $\omega_0$. However, against phase-flips, the qubit is vulnerable to fluctuations in the energy gap. By dressing the qubit, the qubit is rotating about two-axes, at different frequencies, such that both bit and phase-flip errors are protected by an energy gap (see Supplementary Notes 1–3, Supplementary Fig. 1 and Supplementary Movie 1, for further discussion of the theory).

Figure 2a presents a comparison of unprotected Rabi oscillation (red), and a protected-qubit Rabi oscillation using a CCD drive with $\theta_m = 0$, $\phi = 0$ (black), with a close up shown in Fig. 2b. The former is exponentially damped with $T_{Rabi} = 31$ ns. For the CCD scheme, after the first cycle, the amplitude of the Rabi oscillation has stabilized. The stabilized Rabi oscillation is also exponentially damped, but with an extended time of $T_{CCD} = 2.2$ μs. In addition to this amplitude decay, there is also an overall decay of the contrast towards ~ − 5%. This is attributed to $T_1$ relaxation and also affects the conventional Rabi-oscillation, as previously reported for ensembles of NV-centres in diamond[31] (see also Supplementary Note 5 and Supplementary Fig. 4).

In the Fourier domain, the stabilized Rabi oscillation narrows the $\Omega$ frequency component of the signal, and gives rise to two weak sidebands at $\Omega \pm \epsilon_m$. Figure 2c compares the stabilized Rabi oscillation in the case with $\theta_m = 0$, $\pi/2$. For $\theta_m = \pi/2$, the stabilization $\epsilon_m$-term adds in phase with the Rabi-term resulting in an amplitude modulated Rabi drive, and an unprotected Rabi oscillation with strong damping. In the Fourier domain, the phase $\theta_m$ controls the magnitude of the centre band, and the sidebands, see Fig. 2d. Figure 2e, f show maps of the modulation phase ($\theta_m$) and frequency ($\omega_m$) dependence of the signal in the frequency domain with $\theta_m = \pi/2$, respectively. The main feature is a Mollow-triplet[29] with centre frequency $\omega_m$, and sidebands $\omega_m \pm \sqrt{(\omega_m - \Omega)^2 + \epsilon_m^2}$. A feature at $\epsilon_m$ and a counter-rotating term at $2\omega_m$ can also be seen. At high $\omega_m$, additional side bands at $\omega_m \pm \sqrt{(\omega_m - \Omega_{m_I})^2 + \epsilon_m^2}$, $\Omega_{m_I}^2 = \Omega^2 + (m_I A^{nn})^2$, are observed, and attributed to transitions where the nearest neighbour nuclei are in total nuclear spin state $m_I = \pm 1$.

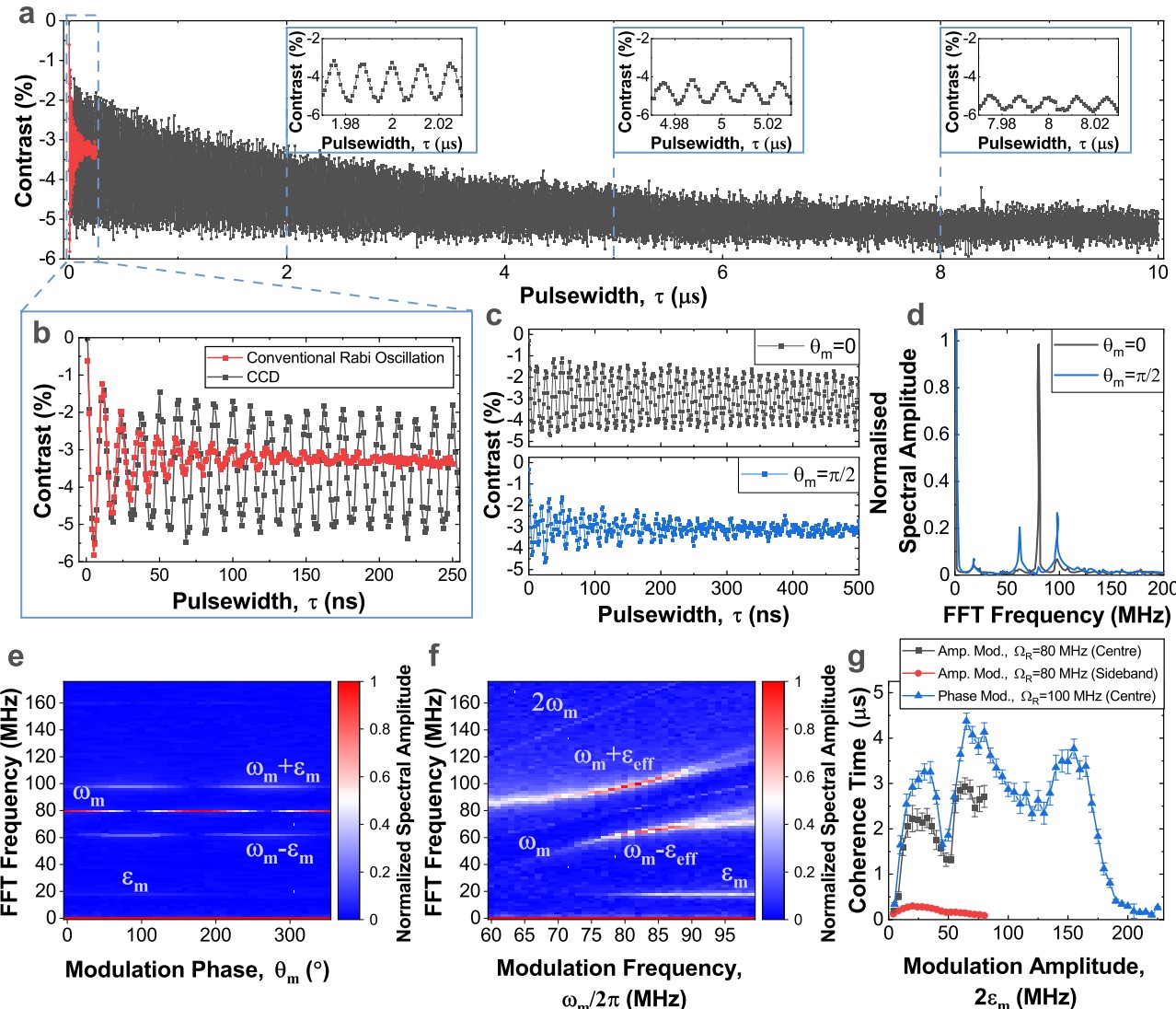

**Fig. 2 | Stabilization of Rabi oscillation. a, b** Comparison of unprotected Rabi oscillation (red) and amplitude modulated CCD scheme with $\epsilon_m = 12$ MHz (grey), showing stabilization of Rabi oscillation on $\mu$s-timescale. **c** Phase control of amplitude modulated CCD scheme. If the modulation phase, $\theta_m = 0$ (grey) the stabilization works, and the spin oscillates at single Rabi frequency. If $\theta_m = \pi/2$ (blue), the $\epsilon_m$-term modulates the amplitude of the Rabi frequency, resulting in a strongly damped oscillation at the two sideband frequencies $\omega_m \pm \epsilon_m$. **d** Fourier transform spectrum of **c. e** Frequency spectrum of amplitude modulated CCD vs modulation phase $\theta_m$, when $\omega_m = \Omega = 80$ MHz and $\epsilon_m = 20$ MHz. $\theta_m$ tunes between single band at $\omega_m$ and two sidebands at $\omega_m \pm \epsilon_m$. **f** Frequency spectrum of amplitude modulated CCD vs modulation frequency $\omega_m$, with $\theta_m = \pi/2$ and $\epsilon_m = 20$ MHz. The main feature is a Mollow-triplet like structure with components $\omega_m$ and $\omega_m \pm \epsilon_m$. **g** Comparison of coherence times of amplitude modulated CCD centre-band (grey) and sidebands (red) and phase modulated CCD centre-band (blue) as a function of modulation amplitude, $\epsilon_m$. The error bars indicate the standard error of the fit coherence times. Source data are provided as a Source Data file.

To characterize the lifetime of the stabilized Rabi oscillation, we plot the coherence time of the centre and sidebands as a function of stabilization field strength $2\epsilon_m$ (Fig. 2g). For an amplitude modulation with $\theta_m = 0$, when $2\epsilon_m > 10$ MHz the energy-gap of the protected qubit exceeds the next-nearest neighbour electron-nuclear hyperfine coupling strengths of ~6.8 MHz[21], and the centre-band coherence time increases to better than a microsecond, as the electron spin is isolated from the majority of nuclear spins. At $2\epsilon_m \approx A = 47$ MHz, there is a dip when the $2\epsilon_m$ sideband-splitting matches the nearest-neighbour hyperfine interaction, before rising to a maximum value of $T_{CCD} \approx 2$ µs. The sidebands are far broader, and the optimum coherence time of the sidebands occurs at $2\epsilon_m \approx 20$ MHz with $T_{CCD}^{\pm} = 0.27$ µs. We interpret the sidebands as providing the stabilization of the Rabi oscillation, and need a faster response to stabilize the electron spin. Using amplitude modulation, $2\epsilon_m$ is limited by the available power.

A similar control can be achieved using phase modulation[42], where the control Hamiltonian is: $H_c = \Omega \cos(\omega t + \phi - \frac{2\epsilon_m}{\Omega}\sin(\omega_m t - \theta_m))S_x$. This has the advantage of not requiring so much power, allowing us to extend the range of $2\epsilon_m$. Qualitatively, the coherence time dependence for phase modulation imitates the amplitude modulation case. However, an additional dip in coherence time is observed at about $100 - 140$ MHz, which corresponds to $2\epsilon_m \approx 2\Omega - A$. When $\epsilon_m > \Omega$ the stabilization stops working. Overall, the phase modulation scheme works better, possibly because the frequency stability of the function generator is better than the power stability, and reaches a maximum of $T_{CCD} = 4.4$ µs, a >150-fold improvement on $T_{Rabi} \approx 25$ ns of the conventional Rabi oscillation.

## Coherent control of protected electron spin
So far, we have shown that at room temperature and milli-Tesla magnetic field, an ensemble of electron spins in boron vacancies decohere

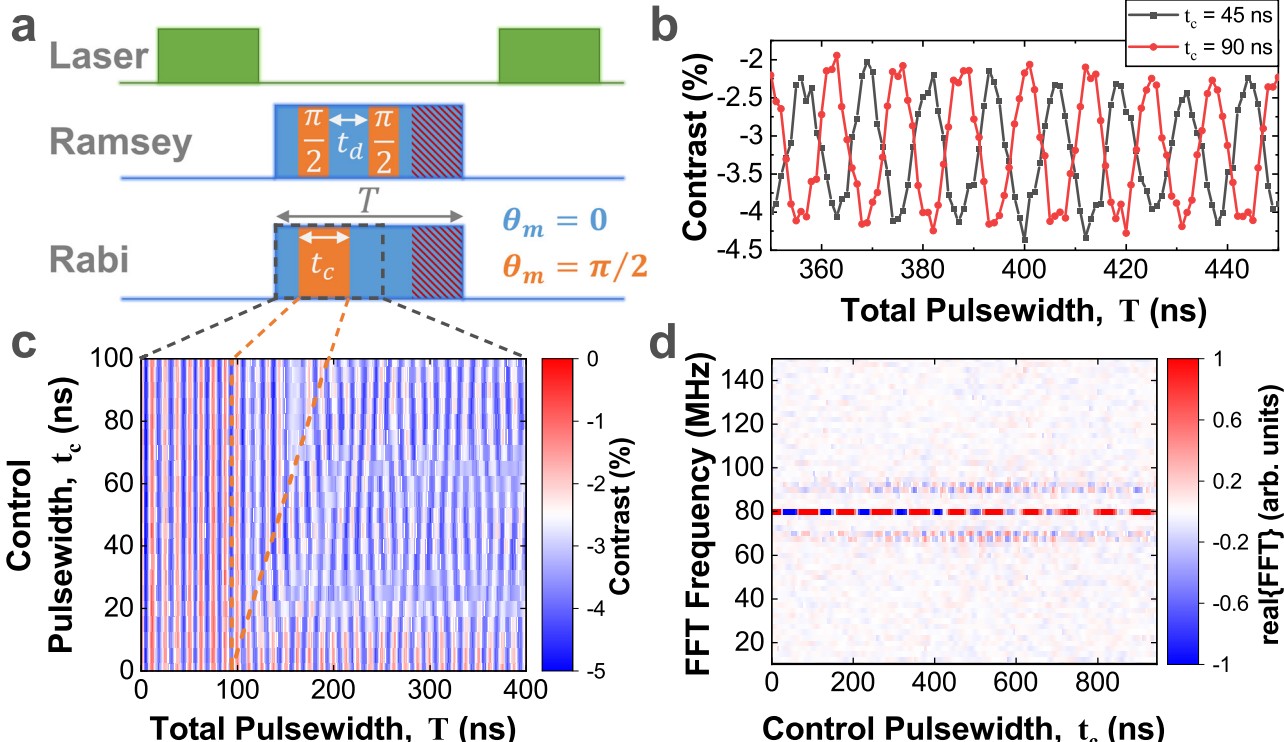

**Fig. 3 | Coherent control of protected qubit. a** To perform a Rabi oscillation of the protected qubit the amplitude modulated CCD pulse is applied for a total time $T$. In the dressed states basis, this results in an effective field of magnitude $\epsilon_m/2$, aligned along polar angles $(\theta_m, \phi_m)$ as given by Eq. (6). For the first 94 ns, the protected qubit is prepared by applying modulation phase, $\theta_m = 0$, initiating a stabilized Rabi oscillation. The control field is tilted by setting $\theta_m = \pi/2$ for a time $t_c$, and then returned to $\theta_m = 0$ to realign the control field along the qubit axis. The qubit state is read out by measuring the phase of the stabilized Rabi oscillation during the time window $t = 1000 - 1400$ ns. **b** Examples of stabilized Rabi oscillations, recorded during the readout window for $\pi$ ($t_c = 45$ ns, grey) and $2\pi$ ($t_c = 90$ ns, red) control pulses. **c** Measured contrast as a function of total pulsewidth, $T$ and control pulsewidth, $t_c$. **d** Real part of FFT of the readout window in **c**. Source data are provided as a Source Data file.

on a timescale of $T_{echo} \approx 100$ ns, due to a strong electron-nuclear hyperfine interaction. However, by using a CCD scheme it is possible to stabilize the Rabi oscillation of the electron spin, increasing the damping time of the Rabi oscillation up to several µs. Regarding applications in metrology or as a spin qubit, the question is, do we simply lock the spin to a Rabi oscillation, or can we define a fully controllable protected qubit, and if so, how coherent is that qubit?

To address this question, we define a protected qubit basis in the second rotating frame. When the drive is set to 'idle' with $\theta_m = \phi = 0$, and $\Omega = \omega_m$, the effective field $H''_c = \frac{\epsilon_m}{2}\sigma''_z$ points along the $z''$-axis, see eqn (6). The protected qubit states are defined as $|0''\rangle = \cos(\omega_m t/2)|0'\rangle - i\sin(\omega_m t/2)|1'\rangle$, and $|1''\rangle = i\sin(\omega_m t/2)|0'\rangle + \cos(\omega_m t/2)|1'\rangle$. In the lab frame, these states are observed as a Rabi oscillation that is in/out of phase with a clock rotation in $S_z$ that starts at $t = 0$, when the drive is turned on, and has a frequency $\omega_m$. If the electron spin is in a superposition state, this will show up as modulation sidebands with frequency component $\Omega \pm \epsilon_m$.

To measure a Rabi oscillation in the protected basis, the experiment has initialization, control and read-out steps, as illustrated in Fig. 3a. To initialize the spin in state $|0''\rangle$, the drive is turned on in the 'idle' state, and left for 94 ns. This is a time of several periods of the Rabi oscillation giving the system time for the frequency to be well defined. To effect a Rabi oscillation, the control field is tilted into $y''$-direction by setting $\theta_m = \pi/2$ for time 94 ns $< t < t_c + 94$ ns, and then switched back to 'idle'. This causes the spin to rotate about the $y''$-axis for a time $t_c$ creating a superposition of $|0''\rangle, |1''\rangle$. For read-out, the PL contrast is measured as a function of total MW-pulse length $T$, and the resulting state of the protected qubit is inferred from the phase of the Rabi oscillation.

Figure 3b, compares the Rabi oscillation after a control pulses of duration $(t_c = 45$ ns, $\epsilon_m t_c = \pi)$, and $(t_c = 90$ ns, $\epsilon_m t_c = 2\pi)$, and shows that the phase of the Rabi oscillation has been shifted by $\pi$. More generally, we measure the PL contrast as a function of the total MW-pulse width $T$ and the control pulse length $t_c$, see the colour map of Fig. 3c. To analyze this we consider the data-set 1000 ns < $T$ < 1400 ns, and make a Fourier transform with respect to $T$, see Fig. 3d. The centre band oscillates with the control pulse length at a frequency $\epsilon_m$, the sidebands oscillate in quadrature with the centre peak, and appear when the protected spin points along the $y''$-direction. As shown in Fig. 4a, the centre peak decays with a time $T_{pRabi} = 487$ ns.

To show rotation of the protected spin about a second axis, a Ramsey interference experiment is performed, using the pulse sequence shown in Fig. 3a. The MW-pulse is applied for a total time 1 < $T$ < 1.4 µs. Again, to initialize the protected spin in state $|0''\rangle$ the MW-field is applied in the 'idle' state for a short time. To apply a $\pi/2$-pulse, $\theta_m$ is switched to $\theta_m = \pi/2$ for a time $\tau_{90} = \pi/2\epsilon_m$, and then switched back to the $\theta_m = 0$ idle state. A pair of $\pi/2$-pulses with a time-delay $t_d$ are applied, and the final state is deduced by analyzing the signal as the total time $T$ and the time-delay $t_d$ are varied, see Fig. 4e. A Ramsey interference showing rotation about $z''$-axis with a frequency of $\epsilon_m$ is observed, with a $T_{pRam} = 706$ ns, demonstrating full control of the protected spin.

To evaluate the potential coherence time of the protected spin, we measure the damping time of the protected Rabi oscillation as a function of the modulation $2\epsilon_m$, see Fig. 4c. The best coherence time $T_{pRabi} = 0.8$ µs is observed at low $\epsilon_m$, which is ~8 times better than the unprotected spin echo time.

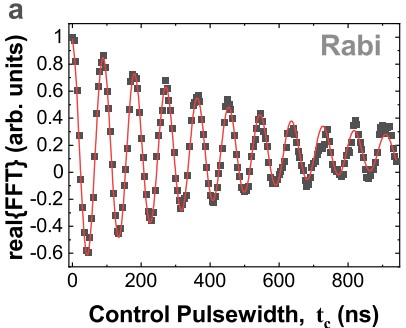

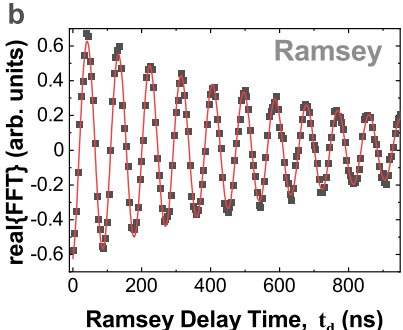

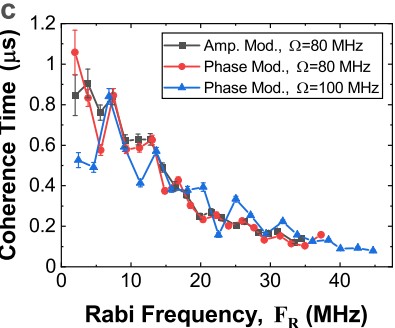

**Fig. 4 | Full control and coherence of the protected qubit. a** The centre-band of the Mollow-triplet oscillates at protected Protected Rabi frequency $F_R \approx \epsilon_m$. **b** Ramsey interference of protected qubit, using measurement protocol shown in Fig. 3a. **c** Rabi coherence time of the protected qubit for different pulse schemes and control field amplitudes. The coherence times are best at low protected Rabi frequency, with maximum times of about 800–1000 ns. The error bars indicate the standard error of the fit coherence times. Source data are provided as a Source Data file.

## Discussion

To conclude, the spin echo coherence time of ensembles of boron vacancies in room temperature hexagonal boron nitride is limited by electron-nuclear interactions to under 100 ns at sub 100 mT B-fields. To overcome this issue, we trial a CCD scheme to stabilize the Rabi oscillation, extending the Rabi damping time up to $T_{CCD} = 4.4\ \mu s = 0.44 T_1$. This time is similar to the spin echo coherence times of InAs/GaAs quantum dots at helium temperatures[43,44], and NV-centres in 10–35 nm diameter nanodiamonds at room temperature[45], and is close to the limit of $0.5 T_1$, reported for NV-centres in diamond[46] where $T_1$ is limited by two-phonon induced spin relaxation. We note that Gottscholl et al.[16] report a temperature$^{-2.5}$ dependence of $T_1$. This is close to the exponent of $-3$ expected for two-phonon induced spin relaxation in the case of a 2D phonon bath. Therefore a similar limit may apply for the case of $S = 1$ boron vacancy in hBN. In that case, reducing the temperature should improve $T_{CCD}$. Further improvements could also be made by adding higher order drive terms[25].

Furthermore, we define a protected qubit basis, show two-axis control, and show that the arbitrary superpositions of the protected qubit can survive for up to 800 ns. This demonstrates that the protection scheme is compatible with applications of the spin as a quantum memory, and with ac B-field sensing schemes[42]. The method avoids the use of cryogenic superconducting magnets, and is therefore suitable for room temperature applications. It should be insensitive to the details of the nuclear bath, and applicable to other spin defects in hBN or other III-V materials with harsh nuclear spin environments.

## Methods

### Sample

The sample consists of a chromium/gold (20/170 nm thick) coplanar waveguide (CPW), with a 10 μm wide central conductor, on a sapphire substrate. An hBN flake, ~100 nm thick, is placed on top of the CPW using the PDMS transfer method. Boron vacancies are generated/activated using C ion irradiation with an energy of 10 keV and dose of $1 \times 10^{14}\ cm^{-2}$. Further details can be found in Baber et al.[17].

### Experimental setup

Photoluminescence is excited using a 532 nm diode-pumped solid-state laser, modulated by an acousto-optic modulator. The laser is coupled to a long working distance objective lens (N.A. = 0.8) which focuses the light to a diffraction-limited spot ~1 μm in diameter. The luminescence is collected with the same objective via a 750 nm long pass filter, to a fibre coupled single photon avalanche diode (SPAD). The intensity is recorded using a time-correlated single photon counting module. The microwave waveforms are generated using an arbitrary waveform generator, amplified (30 dB amplification,

maximum output power 30 dBm) and applied to the CPW on the sample. The optical and microwave excitation and photon collection are synchronised using a digital pattern generator.

## Data availability

Source data are provided with this paper. All other data that supports the findings of this study are available from the corresponding author upon reasonable request. Source data are provided with this paper.

## Code availability

The codes used for the analysis included in the current study are available from the corresponding author upon reasonable request.

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

## Acknowledgements

This work was supported by the Engineering and Physical Sciences Research Council [Grant numbers EP/S001557/1 (I.J.L.), EP/L015331/1 (A.J.B.) and EP/T017813/1 (A.J.B.)] and Partnership Resource Funding from the Quantum Computing and Simulation Hub [EP/T001062/1 (I.J.L. and A.J.B.)]. Ion implantation was performed by Keith Heasman and Julian Fletcher at the University of Surrey Ion Beam Centre. We thank Dr J. P. Hadden for useful discussions at an early stage of the project, and James Haigh for help with the SI Movies.

## Author contributions

I.J.L. and A.J.R. conceived and designed the experiments. S.B. fabricated the sample. I.J.L, R.H. and C.J.P. built the experimental setup and performed the measurements. I.J.L., A.J.R. and A.J.B. supervised the project. I.J.L., C.J.P., A.J.B., R.H. and A.J.R. analysed and discussed the experimental results. A.J.R. and D.R.M.A.-S. performed calculations. A.J.R. wrote the manuscript with contributions from I.J.L and A.J.B.

## Competing interests

The authors declare no competing interests.
