## [Peer Review File · Nature Communications]

REVIEWER COMMENTS

Reviewer #1 (Remarks to the Author):

Ramsay and colleagues reported an experimental study of coherence protection of boron vacancy (VB-) spins in h-BN. They have employed the continuous concatenated driving (CCD) method to engineer the spin coherence property of the VB- spin ensemble. The CCD method is an established method that has been successfully developed and applied to other well-known solid-state spin qubits, such as the diamond NV center. The authors have shown that the CCD method works well for the VB- spin in h-BN to enhance the spin coherence. The authors also demonstrated the Rabi and Ramsey interference experiment in the protected spin space, suggesting that the scheme is compatible with a quantum sensing application.

Quantum spin defects in h-BN are emerging spin qubit systems, which hold great promise for solid-state quantum information. However, several recent studies reported that the spin coherence time of the VB- spin in h-BN is very short, which may significantly limit the potential of the spin as qubit candidates. Thus, the study of Ramsay and their co-workers is timely and it may have a significant impact on the literature. I found, however, that there are a number of major issues to be addressed before I decide to recommend its publication.

It seems to me that the theory behind their method and discussion is poorly described. I would suggest that the authors provide Supplementary Information to present a detailed theoretical description of their method and to assist their discussion of the main result.

In addition, I have a few other related suggestions:

1) Did the authors adopt an existing CCD method (e.g. their reference 20 - 26) without any modification? As correctly pointed out by the authors, physics governing the decoherence of VB- in h-BN is different from those for III-V quantum dots or for defects in group-IV materials. So, I am wondering if this difference affects their implementation of the CCD method.

2) At the end of page 5, the authors stated that "The second term applies a corrective rotation about the y-axis. This is maximum ... it will rotate the spin closer to the target value. This results in a kind of spin-locking effect that can sustain the Rabi oscillation." This explanation is hard to understand and unclear. It would be very useful to provide a proper theoretical guide or visualization for these explanations, especially for readers who are not familiar with the CCD method. In addition, why is it

"kind of" spin-locking? Can the authors specify which is similar to or which is different from the conventional spin locking?

3) A similar issue follows on the next page from line 185 to 190: "In the lab frame, the protected states ... this induces an energy gap e_m over the full Bloch-vector trajectory of the Rabi oscillation". Can the authors provide a detailed theoretical guide or visualization of this statement in Supplementary Information?

The authors stated that the CCD method works extremely well in h-BN. Would it be possible to discuss its theoretical limit or its performance in comparison to other previous relevant studies?

Also, would there be any room for improvement to further protect the spin coherence in h-BN? Please add a discussion on this if there is any. This type of discussion would be useful for future studies.

Figure 3a is hard to read. In particular, yellow-color text and white text in the yellow bars are hard to read. The middle panel seems to present a contrast from blue to red. But, it seems to me just almost all blue, and hard to see what significant information is contained. Is there any way to improve the figure?

The first paragraph of page 9 (line 277 to line 300) is hard to follow as it contains lots of information in a condensed manner. I would suggest splitting the paragraph and adding a supplementary explanation to ease the reading.

Reviewer #2 (Remarks to the Author):

The manuscript presents a systematic study of microwave-induced Rabi oscillations of the VB- spin centre in hBN. For the first time, the CCD scheme was applied to VB- defects reported for NV centres in diamonds, including the observation of a Mollow triplet. Although the content of the paper is complex and is probably more suitable for experts with a strong background in spin resonance, it addresses new, previously unpublished aspects of these new spin defects in a van der Waals material like hBN.

Below are some comments that need to be addressed.

The presented ODMR spectrum with unresolved HF structure indicates a very high applied microwave power, on the other hand the authors mention that the “dephasing rate is proportional to the Rabi frequency possibly due to fluctuations in the power of the microwave source”. Did the authors perform experiments with lower powers? What is B1 field applied?

Like the arXiv manuscript (Ref. 6), the authors concluded that most of the electron spin coherence is lost in the first 100 ns and hypothesised that a small fraction of the coherence might remain on a longer time scale. The authors terminated their spin echo (SE) measurements at 250 ns. Why so early? As contradictions between previous SE experiments in Ref. 13 are highlighted, it would be interesting to see what happens at longer times, e.g. if any ESEEM features are seen.

The ZFS E parameter of 150 MHz is much larger than 49 MHz in Ref. 5 and also than 126 MHz in an earlier work by the same team, although the D value and PL are identical. In both cases, carbon irradiated samples were studied. Is there an explanation for this strong deviation? I am curious to see the ODMR spectrum at B=0.

Can this CCD protocol be called spin coherence or is the term spin-locking more correct? In this case, I suggest mentioning this fact in the title.

Minor remark:

I am not sure if it is fair to discuss the irrelevance of literature data on spin coherence obtained using superconducting magnet at this stage of hBN spin defect research, instead of discussing the reasons for the difference, also taking into account the complexity of the applied protocol and the experimental conditions in the present work.

Summarising, I am generally positive about this paper, although I would recommend making the paper more accessible to a broader readership, particularly in a second half.

The authors are requested to address the above concerns prior to further consideration.

Reviewer #3 (Remarks to the Author):

This manuscript reports on the coherent manipulation of ensemble of negatively charged boron vacancies in hexagonal Boron Nitride. The experiments are realized on a 100-nm thick hBN flake positioned on a CPW, at room temperature and low magnetic field.

The authors convincingly study the spin properties of the Boron vacancy color centers. For that, they have utilized techniques mostly based on pulsed optically detected magnetic resonance. By applying concatenated continuous field drives (CCD), the authors show an increase of the Rabi damping time above 4 μ s (150-fold higher than without CCD). Then, the authors perform a Ramsey experiment, based on a spin evolution in the equatorial plane of the Bloch sphere.

Although CCD has already been demonstrated in the literature, the successful demonstration of a protected ensemble of qubit in a III-V material, where all the host atoms have a finite nuclear spin, is the principal achievement of this work. It differs from Ladd et al, PRB 71, 014401 where the studied spins were nuclear spins, individually much less coupled to the ^{29}Si host nuclear spin bath. It also avoids using a DNP pre-sequence (Bluhm et al., Nature Phys. 7, 109), post selection by continuously estimating the Overhauser field, as demonstrated in GaAs quantum dots by Chekhovich et al., Nature Mat. 12, 494, or applying a B-field gradient (Tyryshkin et al., Nature Mat. 11, 143) to reach longer coherence times.

However, there are few points that are unclear yet:

1. The authors justify using ODMR on hBN to study the spin properties of color centers. They make a point that identifying a bright color center with a long-lived spin coherence time will have high impact, on quantum memory and B-field sensing. However, they also state that Boron vacancies suffer a low brightness. Could that point be clarified? Towards making B-field sensing, I would expect more discussions about the possibility to isolate/make a single Boron vacancy center. For quantum memory applications, could the authors clarify what storage schemes are envisioned with Boron vacancies (single or ensemble of centers)?
2. As I understand, the application of a CCD will prevent (or average out) the effect of spectral diffusion caused by the nuclear spin bath. However, the authors employed a carbon-ion implantation with a dose of $1 \times 10^{14} \text{ cm}^{-2}$, which potentially leads to a very high concentration of vacancy centers. One may therefore expect an interaction of a Boron vacancy center with nearby either resonant or off-resonant centers. Can the author comment on the potential effect of the high spin concentration on the resulting coherence properties under CCD? What could be expected when lowering the defect concentration (targeting single center operations)?
3. In Fig. 2a, the authors compare the unprotected Rabi oscillation with the result of an amplitude modulated CCD. As shown in b, the unprotected method leads to a dephasing at long τ (readout value at 50% of the total contrast). In contrast, the amplitude modulated CCD results tend to a finite -5% contrast, which I believe corresponds to a $m_s \neq 0$ spin state (not necessarily $m_s = +1$). Could the authors comment on that last point? Should we assume that the CCD modulation implies a spin relaxation to the $m_s = 1$ state? Or is it that the spin state goes out of the $\{m_s = 0, m_s = 1\}$ space? Did the authors perform a more complete state tomography to check that point?

This work is timely in a competitive field of research. I will recommend publication if the authors can clarify these points. Please find below more detailed feedback on dedicated points of the paper.

I also have more minor questions and comments about the manuscript.

4. Towards using hBN for B-field

5. Fig. 3a is hardly readable.

6. Can you comment on the value 94 ns used to prepare the protected state?

7. Concerning the application of a static magnetic field, would the co-integration of an hBN foil with a nanofabricated magnet envisioned?

8. Is there a protocol to perform single-shot readout of a protected spin?

9. Would a coplanar microwave resonator instead of a CPW help reaching high Rabi frequencies?

REVIEWER COMMENTS

Reviewer #1 (Remarks to the Author):

Ramsay and colleagues reported an experimental study of coherence protection of boron vacancy (VB-) spins in h-BN. They have employed the continuous concatenated driving (CCD) method to engineer the spin coherence property of the VB- spin ensemble. The CCD method is an established method that has been successfully developed and applied to other well-known solid-state spin qubits, such as the diamond NV center. The authors have shown that the CCD method works well for the VB- spin in h-BN to enhance the spin coherence. The authors also demonstrated the Rabi and Ramsey interference experiment in the protected spin space, suggesting that the scheme is compatible with a quantum sensing application.

Quantum spin defects in h-BN are emerging spin qubit systems, which hold great promise for solid-state quantum information. However, several recent studies reported that the spin coherence time of the VB- spin in h-BN is very short, which may significantly limit the potential of the spin as qubit candidates. Thus, the study of Ramsay and their co-workers is timely and it may have a significant impact on the literature. I found, however, that there are a number of major issues to be addressed before I decide to recommend its publication.

It seems to me that the theory behind their method and discussion is poorly described. I would suggest that the authors provide Supplementary Information to present a detailed theoretical description of their method and to assist their discussion of the main result.

(R1.0) To better describe the theory behind the method, a Supplementary Information file and movie have been prepared. This includes Supplementary Notes 1 to 3, which cover the theory of continuous concatenated decoupling; the locking of the Rabi frequency to the modulation frequency; and the decoupling of the qubit from noise.

In addition, I have a few other related suggestions:

(C1.1) Did the authors adopt an existing CCD method (e.g. their reference 20 - 26) without any modification? As correctly pointed out by the authors, physics governing the decoherence of VB- in h-BN is different from those for III-V quantum dots or for defects in group-IV materials. So, I am wondering if this difference affects their implementation of the CCD method.

(R1.1) For the data presented in Fig. 2, the CCD method is the same as used for NV-centre in diamond used in refs. [24-31]. However, we use a stronger Rabi drive of 80 to 100 MHz to overcome the larger nearest-neighbour hyperfine interaction of 47 MHz. Typically for NV-centres in diamond, a Rabi frequency of about 4-5 MHz is used [28]. For the experiments in fig. 3 and 4, we present an extension of the method to allow arbitrary rotation of the protected spin state.

To clarify this point, we have modified the text at lines 70-73, which now reads:

“This method has previously been used in NV-centres in diamond (24-31), but here we show that it works extremely well in a III-V material with hostile nuclear environment, by using a Rabi drive stronger than the hyperfine coupling.”

(C1.2) At the end of page 5, the authors stated that "The second term applies a corrective rotation about the y-axis. This is maximum ... it will rotate the spin closer to the target value. This results in a kind of spin-locking effect that can sustain the Rabi oscillation." This explanation is hard to understand and unclear. It would be very useful to provide a proper theoretical guide or visualization for these explanations, especially for readers who are not familiar with the CCD method. In addition, why is it "kind of" spin-locking? Can the authors specify which is similar to or which is different from the conventional spin locking?

(R1.2) As part of the rewrite, this section of the text has been cut and the discussion shifted to the next paragraph, see (R1.3). A detailed theoretical guide has been added as supplementary notes 1 to 3. To assist with visualisation of the explanation will have prepared a movie simulation of the evolution of spins under the action of CCD, included with the SI and discussed in Supplementary Note 2.

On rewriting, the term "spin-locking" has been removed because it is not necessary. As we understand, spin-locking refers to locking the spin in a fixed orientation. Here, the spin is locked to a rotating state.

(C1.3) A similar issue follows on the next page from line 185 to 190: "In the lab frame, the protected states ... this induces an energy gap e_m over the full Bloch-vector trajectory of the Rabi oscillation". Can the authors provide a detailed theoretical guide or visualization of this statement in Supplementary Information?

(R1.3) This paragraph has been rewritten as follows (lines 188 to 202).

"In the second rotating frame, the control field is equivalent to a d.c. magnetic field of magnitude ε_m that is oriented using the phases of the drive θ_m and ϕ . By setting the drive such that $\Omega = \omega_m$, $\theta_m = \phi = 0$, the Hamiltonian reduces to $H_C'' = \frac{1}{2} \varepsilon_m \sigma_z''$. In this frame, an error in the Rabi frequency ($\Omega - \omega_m$) results in a rotation about the x'' -axis, and a spread in Rabi frequencies leads to a divergence in the rotation angle, and a damping of the unprotected Rabi oscillation. However, if an ε_m -drive is applied, an error in the Rabi frequency results in a small tilt in the effective magnetic field, limiting the spread in the Bloch vector, resulting in a long-lived Rabi oscillation.

For a spin with no drive, the qubit is protected from bit-flip errors by an energy gap ω_0 . However, against phase-flips, the qubit is vulnerable to fluctuations in the energy gap. By dressing the qubit, the qubit is rotating about two-axes, at different frequencies, such that both bit and phase-flip errors are protected by an energy gap. See Supplementary Notes 1 to 3, for further discussion of the theory."

We have also added a detailed theoretical guide as Supplementary notes 1 to 3, and a Supplementary Movie.

(C1.4) The authors stated that the CCD method works extremely well in h-BN. Would it be possible to discuss its theoretical limit or its performance in comparison to other previous relevant studies? Also, would there be any room for improvement to further protect the spin coherence in h-BN? Please add a discussion on this if there is any. This type of discussion would be useful for future studies.

(R1.4) To address this comment we have added to the discussion section. In particular, the text at lines 343 to 355 is as follows:

“To overcome this issue, we trial a CCD scheme to stabilize the Rabi oscillation, extending the Rabi damping time up to $T_{CCD} = 4.4 \mu s = 0.44T_1$. This time is similar to the spin echo coherence times of InAs/GaAs quantum dots at helium temperatures [43,34], and NV-centres in 10-35 nm diameter nanodiamonds at room temperature [45], and is close to the limit of $0.5T_1$, reported for NV-centres in diamond [42] where T_1 is limited by two-phonon induced spin relaxation. We note that Gottscholl *et al* (16) report a temperature^{-2.5} dependence of T_1 . This is close to the exponent of -3 expected for two-phonon induced spin relaxation in the case of a 2D phonon bath. Therefore, a similar limit may apply for the case of S=1 boron vacancy in hBN. In that case, reducing the temperature should improve T_{CCD} . Further improvements could also be made by adding higher order drive terms [25]. “

Some examples of the performance of CCD in various diamond NV centre experiments is presented in table below.

Reference	T_{CCD} (μs)	T_1 (μs)	Ratio	Material
This work (hBN)	4.4	10	0.44	V_B^- in hBN
Cao et al (28)(NV@diamond)	31	87	0.36	NV in nanodiamond
Stark, (27)	1430	3000	0.48	Single NV in isotopically purified diamond
Wang et al. (24)	500	2400	0.2	Ensemble of NVs in isotopically purified diamond

(C1.5) Figure 3a is hard to read. In particular, yellow-color text and white text in the yellow bars are hard to read. The middle panel seems to present a contrast from blue to red. But, it seems to me just almost all blue, and hard to see what significant information is contained. Is there any way to improve the figure?

(R1.5) Figure 3 has been split into two figures, now labelled Fig. 3 and Fig. 4. A zoom in of the middle panel, now Fig. 3b, is used to give a better view of the data. The colour scheme has been modified to be clearer.

(C1.6) The first paragraph of page 9 (line 277 to line 300) is hard to follow as it contains lots of information in a condensed manner. I would suggest splitting the paragraph and adding a supplementary explanation to ease the reading.

(R1.6) We have rewritten and extended this paragraph into three paragraphs (lines 288 to 316).

Reviewer #2 (Remarks to the Author):

The manuscript presents a systematic study of microwave-induced Rabi oscillations of the VB- spin centre in hBN. For the first time, the CCD scheme was applied to VB- defects reported for NV centres in diamonds, including the observation of a Mollow triplet. Although the content of the paper is complex and is probably more suitable for experts with a strong background in spin resonance, it addresses new, previously unpublished aspects of these new spin defects in a van der Waals material like hBN.

Below are some comments that need to be addressed.

(C2.1) The presented ODMR spectrum with unresolved HF structure indicates a very high applied microwave power, on the other hand the authors mention that the “dephasing rate is proportional to the Rabi frequency possibly due to fluctuations in the power of the microwave source”. Did the authors perform experiments with lower powers? What is B1 field applied?

(R2.1) For the CCD measurements a high Rabi drive of 80 MHz ($B_1=2.86$ mT) is used. The unprotected Rabi oscillation is damped with a time $T_{Rabi} = 30$ ns, see Fig. 1f, limited by phase-flip errors due to low frequency fluctuations in the drive power, which get worse at higher powers. In the protected frame, the Rabi drive creates an energy-gap that protects against both phase and bit-flip errors induced by low frequency fluctuations in magnetic field. The cut-off in the noise spectrum is approximately the hyperfine coupling rate, $A=47$ MHz, hence the high Rabi drive used. A more detailed discussion of the noise protection has been added as Supplementary Note 3.

The figure below shows a CWODMR spectrum at lower Rabi drive of 1MHz ($B_1=36$ μ T), where HF structure is not resolved.

Figure R1: CWODMR spectrum at lower Rabi drive of $\Omega=1$ MHz, ($B_1=36$ μ T).

(C2.2) Like the arXiv manuscript (Ref. 6), the authors concluded that most of the electron spin coherence is lost in the first 100 ns and hypothesised that a small fraction of the coherence might remain on a longer time scale. The authors terminated their spin echo (SE) measurements at 250 ns.

Why so early? As contradictions between previous SE experiments in Ref. 13 are highlighted, it would be interesting to see what happens at longer times, e.g. if any ESEEM features are seen.

(R2.2) At the time of performing the experiment, it was natural to terminate the measurement after 250 ns, since the electron spin coherence had dropped below the noise floor. To reconcile our observations with ref. (16 [ref. 13 in previous manuscript]), we did perform measurements to longer times. Whilst we observe ESEEM feature at short times see Fig. 1f, we do not observe a long-lived tail (see Fig. R2). In principle, the signal to noise is good enough to resolve a long-lived tail of similar amplitude to ref. (16). This may be sample related and linked to the unresolved HF structure (see Fig. R1). We have added the data of Figure R2 as Supplementary Figure 3, and the following comment at line 139 of the main paper:

“We could not observe a long-lived tail in the spin echo, see Supplementary Figure. 3.”

Figure R2: (a) Long integration time spin echo measurements with Rabi frequency of 24 MHz. (b) Close-up of the data in (a). This data is included as Supplementary Figure 3

(C2.3) The ZFS E parameter of 150 MHz is much larger than 49 MHz in Ref. 5 and also than 126 MHz in an earlier work by the same team, although the D value and PL are identical. In both cases, carbon irradiated samples were studied. Is there an explanation for this strong deviation? I am curious to see the ODMR spectrum at B=0.

(R2.3) In light of this comment, we decided to measure E again. Previously, in ref. (17) the set up applies a magnetic field by positioning a permanent magnet. By measuring the CWODMR spectra vs B-field, E was deduced from a fit. This was not a good way to do the measurement, since the calibration at low B-field is less reliable.

In the revised measurement, the permanent magnet is removed, and the CWODMR measurement is made in zero B-field. The data is shown in Fig. R3. A fit to

$$\text{Contrast} \propto |\chi_{0+}(f - f_0 - E) + \chi_{0-}(f - f_0 + E)|^2$$

is made. This is the sum of the susceptibilities of the two ESR transitions ($0 \rightarrow \pm 1$) driven by the microwave signal, where $\chi(x) = \frac{\gamma}{\gamma + ix}$. The fit yields E=59 MHz. This value is consistent with the study

of Guo et al [ACS Omega 7 1733 (2022).], where 30 keV N-ion implantation at dose of $1E14 \text{ cm}^{-2}$ yields $E=70 \text{ MHz}$. To address this point, we have added Supplementary Figure 2, and modified the text starting at line 105 as follows:

“.., and $E = 59 \text{ MHz}$ (Supplementary Figure 2), ... E is relatively large, perhaps due to high level of strain caused by use of carbon-irradiation to generate the defects, and is consistent with reports for ion-implanted samples [33].”

Figure R3: CWODMR at zero magnetic field, this data is included as Supplementary Figure 2.

(C2.4) Can this CCD protocol be called spin coherence or is the term spin-locking more correct? In this case, I suggest mentioning this fact in the title.

(R2.4) On rewriting, the term “spin-locking” has been removed because it is not necessary. As we understand, spin-locking refers to locking the spin in a fixed orientation. Here, the spin is locked to a rotating state.

Minor remark:

(C2.5) I am not sure if it is fair to discuss the irrelevance of literature data on spin coherence obtained using superconducting magnet at this stage of hBN spin defect research, instead of discussing the reasons for the difference, also taking into account the complexity of the applied protocol and the experimental conditions in the present work.

(R2.5) The text at line 140 has been changed.

The CCD protocol used is conceptually more complex than a spin echo, but the hardware requirements are similar, and low compared to the cost/complexity of a cryogenically cooled superconducting magnet.

(C2.6) Summarising, I am generally positive about this paper, although I would recommend making the paper more accessible to a broader readership, particularly in a second half.

(R2.6) To address this point, we have followed the suggestions provided by all reviewers, and made substantial improvements to the clarity of the second half of the paper and added a supplement. We thank them for their careful reading of the manuscript, and believe the work is much improved by following their suggestions.

The authors are requested to address the above concerns prior to further consideration.

Reviewer #3 (Remarks to the Author):

This manuscript reports on the coherent manipulation of ensemble of negatively charged boron vacancies in hexagonal Boron Nitride. The experiments are realized on a 100-nm thick hBN flake positioned on a CPW, at room temperature and low magnetic field. The authors convincingly study the spin properties of the Boron vacancy color centers. For that, they have utilized techniques mostly based on pulsed optically detected magnetic resonance. By applying concatenated continuous field drives (CCD), the authors show an increase of the Rabi damping time above 4 μ s (150-fold higher than without CCD). Then, the authors perform a Ramsey experiment, based on a spin evolution in the equatorial plane of the Bloch sphere. Although CCD has already been demonstrated in the literature, the successful demonstration of a protected ensemble of qubit in a III-V material, where all the host atoms have a finite nuclear spin, is the principal achievement of this work. It differs from Ladd et al, PRB 71, 014401 where the studied spins were nuclear spins, individually much less coupled to the ^{29}Si host nuclear spin bath. It also avoids using a DNP pre-sequence (Bluhm et al., Nature Phys. 7, 109), post selection by continuously estimating the Overhauser field, as demonstrated in GaAs quantum dots by Chekhovich et al., Nature Mat. 12, 494, or applying a B-field gradient (Tyryshkin et al., Nature Mat. 11, 143) to reach longer coherence times.

However, there are few points that are unclear yet:

(C3.1). The authors justify using ODMR on hBN to study the spin properties of color centers. They make a point that identifying a bright color center with a long-lived spin coherence time will have high impact, on quantum memory and B-field sensing. However, they also state that Boron vacancies suffer a low brightness. Could that point be clarified? Towards making B-field sensing, I would expect more discussions about the possibility to isolate/make a single Boron vacancy center. For quantum memory applications, could the authors clarify what storage schemes are envisioned with Boron vacancies (single or ensemble of centers)?

(R3.1) An attractive feature of hBN, is that some defects exhibit high brightness [up to 87% internal quantum efficiency in <https://doi.org/10.1364/OPTICA.6.001084>], and sub-10 meV optical emission linewidth at room temperature [exemplified by ref 2, for example]. However, the boron vacancy is not one of these defects. As can be seen in Fig. 1b the PL of VB- spectrum has a width of 100s of meV. The PL signal is not so bright – hence the lack of single boron vacancy studies. In ref. [18], Reimers et al calculate radiative lifetime of 10 μ s, with ns non-radiative recombination. Consistent with this, we have measured PL decay rates of a \sim ns [17]. This suggests the quantum efficiency is low. There is a prospect that other defects in hBN may simultaneously have narrow optical emission and finite ODMR response, such as the recent report of Stern et al [13]. The issue for boron vacancies is that an application as a quantum memory would require both efficient optical read in/out and a long lived spin.

For a B-field sensor with a large ensemble of spins, due to an increase in signal, the sensitivity of a magnetometer scales with square-root of the number of spins [Taylor Nature Phys. 4 810 (2008)] and inversely with the square-root of T_2^* .

$$\eta_{d.c.} \approx \frac{\hbar}{g\mu_B C \sqrt{T_2^*}}$$

However, for very high densities T_2^* is degraded, so for most B-field sensors, there will be an optimum density of spins.

The coherence times of a spin subject to nuclear spin bath is largely a property of the host material [Kanai, PNAS 119 e2121808119 (2022)]. The boron vacancy provides a model system to evaluate the suitability of hBN as a host of spin defects. Recent work [16] suggests that microsecond spin echo times are possible, and this has stimulated considerable interest in spin defects in hBN for potential applications in memory, and B-field sensing. Our work provides a lot of information that allows the reader to make a more informed opinion on the potential of hBN as a host material for these applications.

To address this point, we have modified the introduction in several places as follows:

Lines 34 – 42: “Firstly, ODMR provides a useful tool for identifying defects, and has been instrumental in identifying the boron vacancy in hBN [2, 3]. Secondly, a good magnetic field sensor or spin-photon interface [4] requires a defect with both good optical and spin properties. There is mounting evidence that some hBN defects have excellent optical properties, with high brightness of up to 87% quantum efficiency [5] at visible wavelengths well-matched to silicon detectors [6], and suggestions of transform-limited transitions with a high fraction of emission occurring through the zero-phonon line [7, 8].”

Lines 60-62: “Since nuclear spin dephasing is largely a property of the host material [23], the boron vacancy serves as a model spin system to evaluate the potential of hBN as a host.”

Lines 77 – 84: “The method avoids the use of cryogenic superconducting magnets, and is therefore suitable for room temperature applications. Whilst the boron vacancy may not ultimately be suitable for single spin applications, due to its low quantum efficiency and broad photoluminescence spectrum [17, 18], ODMR on other single spin defects has been reported [13, 14]. The CCD method presented here should be insensitive to the details of the nuclear bath, and therefore applicable in general to spin defects in hBN, and to other III-V materials with harsh nuclear spin environments.”

(C3.2). As I understand, the application of a CCD will prevent (or average out) the effect of spectral diffusion caused by the nuclear spin bath. However, the authors employed a carbon-ion implantation with a dose of $1e14cm^{-2}$, which potentially leads to a very high concentration of vacancy centers. One may therefore expect an interaction of a Boron vacancy center with nearby either resonant or off-resonant centers. Can the author comment on the potential effect of the high spin concentration on the resulting coherence properties under CCD? What could be expected when lowering the defect concentration (targeting single center operations)?

(R3.2) We note that our $T_1=10\ \mu s$ is a bit shorter than ref. (16), where $T_1=20\ \mu s$ was reported. Our value of $E=59\ MHz$ (see R2.3) is also relatively high. In our previous work, ref. [17] a dip in PL at $B=75\ mT$ was reported when the Zeeman splitting of unknown spin $\frac{1}{2}$ defects is resonant with the spin splitting of the boron vacancies, see also ref. [3]. This suggests that the spin concentration is high enough to impact the spin properties of the boron vacancies. Reducing the VB- density may ameliorate this issue, but we are unlikely to be able to target single centre operations, for the reasons discussed in R3.1.

In group IV materials, where nuclear-spin dephasing is relatively weak the coherence times are noticeably reduced at high defect concentration due to dipolar coupling. We note that the spin echo is quite effective at correcting for dephasing due to the dipolar dephasing, and typically if dipolar dephasing is the limiting factor $T_{echo} \gg T_2^*$ [Bauch PRX 8 031025 (2018), Kaspar PRAppl 13 044054 (2020)]. In principle, the magnetic dipole-dipole interaction is relatively material independent, so we

might expect to see the same in hBN. However we observe $T_{echo} \sim T_2^*$, suggesting that dipolar dephasing is not the main source of decoherence.

We note that in ref. (3), Haykal et al also report $T_{echo} < 100$ ns, and observe that T_{echo} depends on the ^{11}B content, confirming that nuclear spins are the source of decoherence on this timescale. The echo times measured are also close to theoretical limits for nuclear spin induced dephasing calculated by Haykal et al in ref. [3]. Therefore, we expect contributions to dephasing from dipolar coupling to be less significant in hBN than in diamond or SiC, and the echo times to be similar at the single defect level.

Provided the effective B-field generated by the dipole fields of nearby defects is weak compared with the Rabi drive of ~ 80 MHz, the CCD should protect the coherence of the spin. We also note that we measure a T_{CCD} of up to $0.44T_1$, and since this is close to the expected limit of $0.5T_1$ (see R1.4) we do not expect to see much change on lowering the defect density.

To more clearly assert that nuclear spins are limiting T_{echo} , we have modified text at lines 132 - 134 as follows:

“These numbers are similar to those of Haykal et al [3], who report that T_{echo} is sensitive to ^{11}B content, and therefore limited by nuclear spins.”

(C3.3). In Fig. 2a, the authors compare the unprotected Rabi oscillation with the result of an amplitude modulated CCD. As shown in b, the unprotected method leads to a dephasing at long tau (readout value at 50% of the total contrast). In contrast, the amplitude modulated CCD results tend to a finite - 5% contrast, which I believe corresponds to a $m_s \neq 0$ spin state (not necessarily $m_s = +1$). Could the authors comment that last point? Should we assume that the CCD modulation implies a spin relaxation to the $m_s = 1$ state? Or is it that the spin state goes out of the $\{m_s = 0, m_s = 1\}$ space? Did the authors perform a more complete state tomography to check that point?

(R3.3) To address this question, we have added supplementary Fig. 4 (reproduced here as Figure R4), and Supplementary Note 6. The overall decay of the contrast towards $\sim 5\%$ is a consequence of the pulse scheme used (Fig. R4a). The readout laser pulse is applied straight after the microwave pulse and therefore the time between the initialization and readout laser pulse varies with the duration of

the microwave pulse. This results in a decay in the signal due to relaxation from a net population in $m_s=0$ (bright).

Figure R4: T_1 decay during CCD. (a) Pulse sequence used for Rabi-oscillations and CCD measurements presented in Fig. 2 of the main text. The dark period, $t_2 = t_{rep} - 3t_1 - 2t_0 - \tau$ decreases as the microwave pulsewidth increases in order to maintain a constant overall repetition period. (b) Comparison between Rabi-oscillations and CCD at pulsewidths approach T_1 . With no microwave applied, the contrast decreases due to the transfer of population from the $m_s = 0$ to $m_s = \pm 1$ states.

In Fig. R4b the measurement of the unprotected Rabi oscillation is extended to longer pulsewidths and shows the same decay as the CCD oscillation. We also add data to show the decay in the signal with no microwaves applied. Hence, the decay is not directly related to the CCD modulation. We did not perform a complete state tomography.

In addition to the supplementary figure and note, we have added the following at lines 222 to 226 of the main text:

“In addition to this amplitude decay, there is also an overall decay of the contrast towards $\sim -5\%$. This is attributed to T_1 relaxation and also affects the conventional Rabi-oscillation, as previously reported for ensembles of NV-centres in diamond[31] (see also Supplementary Note 6 and Supplementary Figure 4).”

This work is timely in a competitive field of research. I will recommend publication if the authors can clarify these points. Please find below more detailed feedback on dedicated points of the paper.

I also have more minor questions and comments about the manuscript.

(C3.4). Towards using hBN for B-field

(R3.4) Due to the nuclear spins, hBN is probably of limited use for the sensing of dc B-fields. We are currently evaluating ac B-field sensing using CCD based on scheme presented in ref. [42].

(C3.5). Fig. 3a is hardly readable.

(R3.5) We have split Fig 3 into two figures, Fig. 3 and 4.

(C3.6). Can you comment on the value 94 ns used to prepare the protected state?

(R3.6) To address this point, we have added the following text at line 297

“To initialize the spin in state $|0\rangle$, the drive is turned on in the ‘idle’ state, and left for 94ns. This is a time of several periods of the Rabi oscillation giving the system time for the frequency to be well defined.”

(C3.7). Concerning the application of a static magnetic field, would the co-integration of an hBN foil with a nanofabricated magnet envisioned?

(R3.7) Examples of co-integration of hBN with a magnetic material have been reported by Huang et al, Nature Comm. 13 5369 (2022), and Kumar, ArXiv 2207.10477 (2022). We now cite these papers in the introduction L48, as refs. (10,11)

(C3.8). Is there a protocol to perform single-shot readout of a protected spin?

(R3.8) Not as far as we are aware. In general, the microwave drive would be switched off at an appropriate time to project the system into the spin up/down basis, and then the procedure for doing standard single shot measurement would ensue.

(C3.9). Would a coplanar microwave resonator instead of a CPW help reaching high Rabi frequencies?

(R3.9) If the bandwidth of the microwaves is not an issue, then one could consider resonator designs to enhance the Rabi drive.

REVIEWERS' COMMENTS

Reviewer #1 (Remarks to the Author):

I found that the issues that I raised were well-addressed. I recommend its publication in Nature communications.

Reviewer #2 (Remarks to the Author):

The authors have addressed all my criticisms, made new measurements, corrected the errors (e.g., E-parameter) and rewritten manuscript in some parts, making it accessible to somewhat broader readership.

Reviewer #3 (Remarks to the Author):

In this updated version, the authors have elaborated on two major elements of the manuscript.

First was the accessibility of the paper. They have added a Supplementary to both describe the details about the theoretical part (Note 1-3) and experimental part (Note 4-5, SuppFig 2-4) of their work. In particular, Note 5 (Note 6 was mentioned in the rebuttal letter) and SuppFig 4 replied to my 3rd point. The authors have also improved the introductory part, making the manuscript more accessible and informative to a broader audience (reply to my 1st point).

Second, they have convincingly replied to my second point about the potential coherence time limitation, and the rewriting of the discussion makes it clearer to understand in the article. Figure 3b is now more readable.

The authors have also responded to other questions and made the corresponding changes. I, therefore, recommend publication in Nature Communications